ns

# Selective sweep probabilities in spatially expanding populations

Alexander Stein [1,2] ✉, Kate Bostock [3], Ramanarayanan Kizhuttil [4,5], Maciej Bak [3] & Robert Noble [3] ✉

Evolution during range expansions shapes biological systems from microbial communities and tumours to invasive species. A fundamental question is whether, when a beneficial mutation arises during a range expansion, it will evade clonal interference and sweep to fixation. However, most theoretical investigations of range expansions have considered regimes in which selective sweeps are effectively impossible, while studies of selective sweeps have assumed constant population size or ignored spatial structure. Here we use mathematical modelling and analysis to investigate selective sweep probabilities and timings in biologically relevant scenarios, including the case in which mutants can displace a slowly spreading wildtype. Assuming constant expansion speed, we find surprisingly simple approximate and exact expressions for sweep probabilities in one, two and three dimensions, which are independent of mutation rate. Agent-based simulations confirm that our predictions are accurate for the spatial Moran process and remain informative when mutation effects on fitness are random and multiplicative. We further compare and synthesise our results with those obtained for alternative growth laws. Parameterised for human tumours, our model predicts that selective sweeps are rare except during early solid tumour growth, thus providing a general, pan-cancer explanation for findings from recent sequencing studies.

Range expansion—the spatial spread of populations into new regions—is ubiquitous across biological scales and alters the course of evolution in distinct, often profound ways that remain incompletely understood[1]. Among cell populations, evolution during range expansions determines the development and spatial heterogeneity of biofilms[2], tumours[3], mosaicism[4] and normal tissue[5]. At the species level, range expansions influenced human evolution[6] and are of growing importance as climate change forces organisms into new habitats[7,8]. Prior theoretical and experimental investigations of evolution during range expansion have considered the case in which the wildtype population spreads into new territory much faster than any mutant can displace the wildtype[9–12]. In this scenario, which is typical of

microbial colonies growing in vitro, mutants essentially expand only at the population boundary and selective sweeps are precluded. The alternative case of slow range expansion and strong selection has been unexplored but is widely plausible. Consider, for example, an invasive species that is rapidly adapting to its new environment while gradually displacing a resident competitor[13], or a bacterial colony whose growth is slowed by sub-inhibitory antibiotic treatment.

Cancer provides an especially strong motivation for investigating the likelihood of selective sweeps during range expansion. Having acquired the necessary driver mutations to escape homeostasis, solid tumours continue to accumulate neutral and driver (that is, cell-fitness-enhancing) mutations as they grow and invade surrounding

[1]Centre for Cancer Evolution, Barts Cancer Institute, Queen Mary University of London, London, UK. [2]Department of Physics, ETH Zurich, Zürich, Switzerland. [3]Department of Mathematics, City St George's, University of London, London, UK. [4]Department of Physics, Indian Institute of Science Education and Research, Kolkata, India. [5]Department of Ecology, Behavior & Evolution, UC San Diego, San Diego, CA, USA. ✉e-mail: alexander.stein@ulb.be; robert.noble@citystgeorges.ac.uk

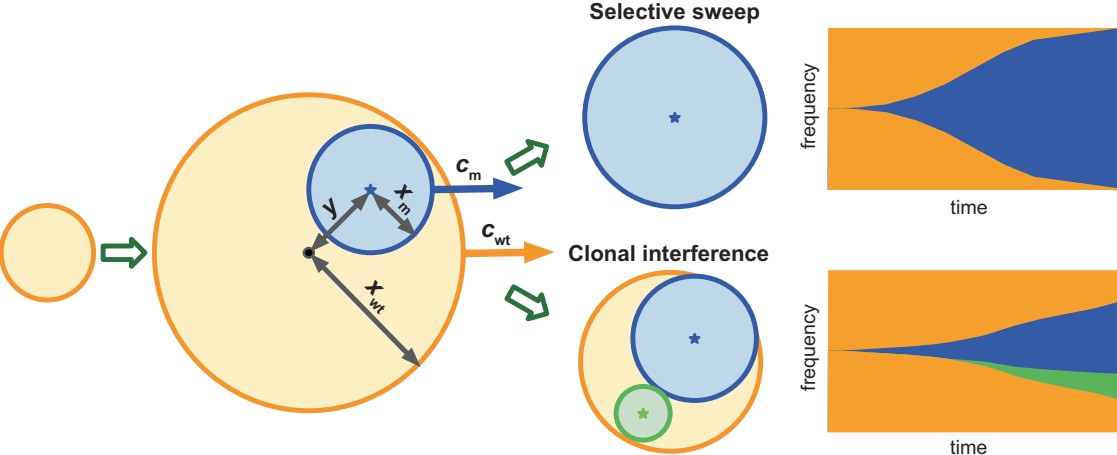

**Fig. 1 | Illustration of the macroscopic model (not to scale) showing the two possible fates of the first surviving mutant (blue) within the wildtype population (orange).** In the first case, the mutant reaches every part of the wildtype boundary before further mutants arise in the wildtype, and so achieves a selective sweep. Otherwise, a second (green) mutant arises in the wildtype population and generates clonal interference.

tissue[14,15]. The mode of this evolutionary process has sparked debate[16,17]. An early, highly influential model based on colorectal cancer genetics posited a linear mode of evolution in which cancers acquire mutations through sequential selective sweeps[18]. Later studies demonstrated intra-tumour heterogeneity with respect to both neutral and driver mutations, suggesting that most cancers undergo branching evolution before they are detected[19–23]. In some cases, the extent of heterogeneity has been shown to predict clinical outcomes[24–26]. A possible explanation that reconciles these observations is that linear evolution is restricted to the very early stages of solid tumour evolution and mutations with very strong selective advantage, yet a systematic study of this general, pan-cancer hypothesis is still lacking[16,17,27]. Mathematical modelling offers a way to investigate the early stages of tumour evolution, which are typically impossible to observe in the clinic[28,29]. We[26,30] and others[31–34] have used models with alternative spatial structures and modes of cell dispersal to examine how spatial structure influences tumour evolutionary mode and the extent of intratumour heterogeneity. However, with notable exceptions[35–38], theoretical investigations have ignored spatial structure[39–41] or relied on agent-based models[26,30–34] whose results can be difficult to interpret, provide only limited explanatory insights and are not readily generalisable.

Here we use mathematical analysis to explain why beneficial mutations typically fixate only−if at all−in the very early stages of range expansions, even when mutants can displace the wildtype faster than the wildtype expands its range. By solving our model in the canonical case of constant radial expansion speed, we derive exact and simple approximate expressions for sweep probabilities in one, two and three dimensions. We confirm the accuracy and robustness of our analytical results using extensive agent-based simulations of a spatial Moran process and we compare outcomes for alternative growth models. We discuss how these findings shed light on the nature of evolution in range expansions in general and cancer development in particular.

## Results

### A macroscopic model of evolution during range expansion
Our macroscopic model is designed to test whether clonal interference alone can prevent selective sweeps and to obtain upper bounds on selective sweep probabilities during range expansions. We consider a wildtype population that starts expanding spherically at time $t = 0$, such that its radius $x_{wt}$ grows at speed $c_{wt}$. Focusing on

selective sweeps, we consider only advantageous mutations, which we assume spread within the wildtype at speed $c_m > c_{wt}$ (Fig. 1). Mutations occur at per-capita rate $\widetilde{\mu}$ with each surviving genetic drift with probability $\rho$. In our analytic model, it is enough to consider the compound parameter $\mu = \rho\widetilde{\mu}$. For brevity, we will refer to $\mu$ as the mutation rate unless otherwise mentioned.

For mathematical and biological reasons (see 'Discussion'), we focus on a model with constant radial expansion speeds; later, we compare these results with those that pertain to alternative growth models. Various models link propagation speeds to fitness values and migration rates. The most prominent formula is obtained from a reaction-diffusion equation associated with Fisher[42] and Kolmogorov[43]: $c = 2\sqrt{aD}$, where $D$ is the diffusion coefficient carrying information of migration and $a$ is the difference in the local proliferation rate. More recent studies have sought to refine and generalise this result[44–47]. Our main results apply to any model that generates approximately constant speeds.

The first surviving mutation achieves a selective sweep only if it reaches every part of the wildtype expansion front before a second mutant of equal or greater fitness arises within the wildtype (Fig. 1). Otherwise, the outcome is clonal interference or possibly a soft selective sweep if the competing mutations are sufficiently similar[48]. For simplicity, in our macroscopic model, we neglect mutants with fitness values between those of the wildtype and the first mutant, which would slow the expansion of the first mutant and so reduce rather than nullify the selective sweep probability. Neither do we investigate the case of a yet-fitter mutant that arises from the first mutant and achieves a selective sweep. Later, we will examine the effects of relaxing these assumptions.

The unconditional sweep probability is derived in four steps. First, we introduce the random variable $X$, the radius of the wildtype population when the first mutant arises and we compute its probability density $f_X(x)$. Second, we introduce the random variable $Y$, the distance between the wildtype and mutant origins and we calculate its probability density conditioned on $X$, namely $f_Y(y|X = x)$. Third, we derive an expression for the conditional sweep probability $\Pr(\text{sweep}|X = x, Y = y)$. Finally, we marginalise out $X$ and $Y$ to obtain the unconditional sweep probability $\Pr(\text{sweep})$. We focus on the three-dimensional case; analogous results in one and two dimensions are presented in the SI Text Sections 5 and 6.

The following result (proved in SI Text Section 1) will be useful in various parts of subsequent derivations.

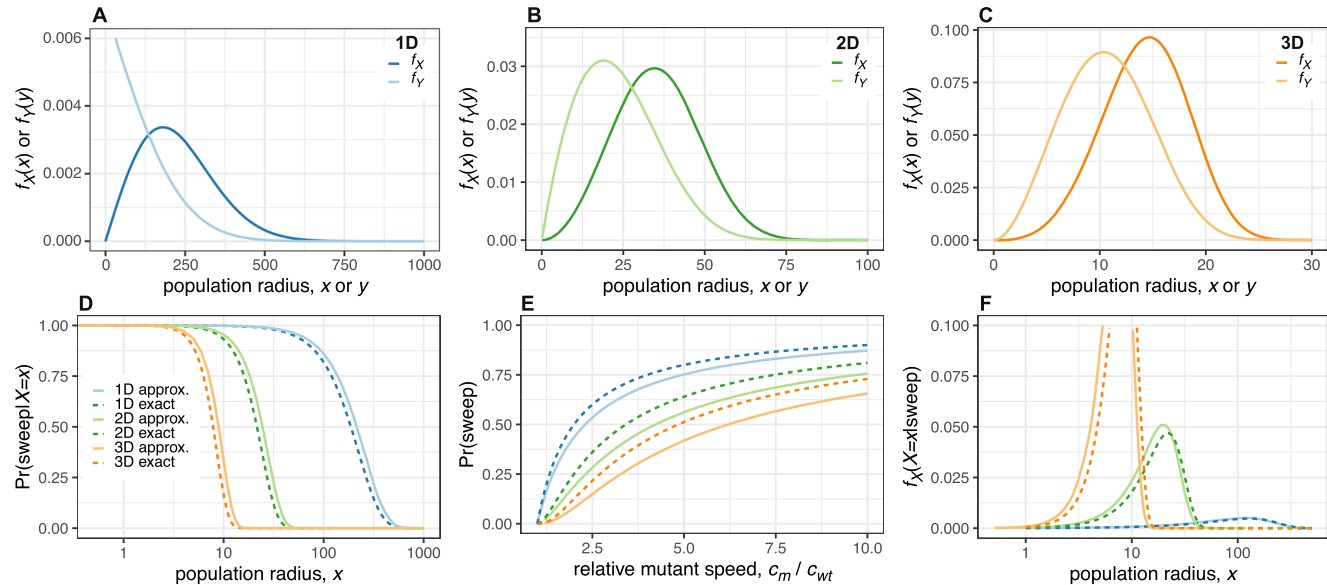

**Fig. 2 | Analytical and numerical solutions of the macroscopic model.** Probability density of the wildtype population radius at the time the first surviving mutant arises (dark curves) and of the distance between the origins of the mutant and wildtype expansions (light curves) in one dimension (**A**), two dimensions (**B**) and three dimensions (**C**). **D** Approximate and exact solutions for the sweep probability conditioned on the wildtype population radius. **E** Approximate and exact solutions for the unconditional sweep probability. **F** Approximate and exact solutions for the probability density of the wildtype population radius at the time the first surviving mutant arises, conditioned on this mutant achieving a selective sweep. Except where parameter values are explicitly varied, we set $c_{\mathrm{wt}} = 0.15$, $\widetilde{\mu} = 10^{-5}$, $\rho = 0.23$ and $c_{\mathrm{m}} = 0.31$. Formulas for all curves are summarised in Supplementary Table S1.

**Claim 1.** The probability that $k$ mutations arise and survive during the time interval $[0, t]$ is Poisson distributed,

$$P_k = \frac{e^{-\lambda}\lambda^k}{k!} \text{ with } \lambda = \mu \int_0^t N(s)\, ds, \tag{1}$$

where $N(s)$ is the population size at time $s$. In particular, the probability that no successful mutant arises is $P_0 = e^{-\lambda}$.

Although it is commonly assumed that mutations are coupled to divisions, it is straightforward to translate between per-capita and per-division mutation rates (see SI Text Section 2).

**Arrival time of the first mutant**

In the absence of mutants, the wildtype population in three dimensions grows as $N_{\mathrm{wt}} = \frac{4}{3}\pi x_{\mathrm{wt}}^3 = \frac{4}{3}\pi(c_{\mathrm{wt}}t)^3$. Applying Claim 1, we obtain the probability that no mutations arise in the time interval $[0, t]$,

$$P_0 = e^{-\mu \int_0^t \frac{4}{3}\pi(c_{\mathrm{wt}}s)^3 ds} = e^{-\left(\frac{t}{\kappa}\right)^4}, \tag{2}$$

in terms of a characteristic duration $\kappa = \sqrt[4]{\frac{3}{\pi\mu c_{\mathrm{wt}}^3}}$. We identify $1 - P_0$ as the cumulative distribution function for the arrival time $T$ of the first surviving mutant. The probability density of $T$ is then

$$f_T(t) = \frac{d(1 - P_0)}{dt} = \frac{4t^3}{\kappa^4}e^{-\left(\frac{t}{\kappa}\right)^4}, \tag{3}$$

which is the Weibull distribution with shape parameter 4 and scale parameter $\kappa$. Substituting $x = c_{\mathrm{wt}}t$, we find that $f_X(x)$, the probability density of the radius of the wildtype population $X$ at the time the first surviving mutant arises, is the Weibull distribution with shape parameter 4 and scale parameter $\theta = \sqrt[4]{\frac{3c_{\mathrm{wt}}}{\pi\mu}}$. It follows that $\mathbb{E}[X] \approx 0.91\,\theta$ and $\mathrm{Var}[X] \approx 0.065\,\theta^2$. Analogous calculations in one and two dimensions yield similar Weibull distributions (SI Text Sections 5 and 6 and Fig. 2A–C).

**Location of the first mutant**

Next, we compute the probability density for the distance $Y$ between the first surviving mutant and the centre of the wildtype population. Since mutants arise in proportion to the number of wildtype cells, we have $f_Y(y|X = x)\, dy \propto D(y)$, where $D(y)$ is the number of cells at distance $y$. $D(y)$ corresponds to the infinitesimal shell, $D(y) = 4\pi y^2\, dy\, \mathbf{1}_{[0, x]}(y)$, where the last term is an indicator function that defines the boundary of the wildtype population. After normalisation, we obtain

$$f_Y(y|X = x) = \frac{3y^2}{x^3}\mathbf{1}_{[0, x]}(y). \tag{4}$$

We calculate the unconditional probability density of $Y$ by marginalising out $X$,

$$f_Y(y) = \int_0^\infty f_Y(y|X = x)f_X(x)\, dx = \frac{3y^2}{\theta^3}\Gamma\left(\frac{1}{4}, \frac{y^4}{\theta^4}\right), \tag{5}$$

where $\Gamma(a, z) = \int_z^\infty t^{a-1}e^{-t}\, dt$ is the incomplete gamma function. We then find $\mathbb{E}[Y] \approx 0.68\,\theta$ and $\mathrm{Var}[Y] \approx 0.070\,\theta^2$. Similar results pertain in one and two dimensions (SI Text Sections 5 and 6; Fig. 2A–C).

**Conditional sweep probability**

To compute the sweep probability, we first need an expression for the remaining wildtype population, $N_{\mathrm{wt}}$. Therefore, we introduce time measure $\tau$, with $\tau = 0$ when the first mutant arises. Recall that $t = 0$ at the origin of the wildtype population, so $t = \tau + x/c_{\mathrm{wt}}$. Once the mutant population starts expanding, we have $N_{\mathrm{wt}}(\tau) = \widetilde{N}_{\mathrm{wt}}(\tau) - \Delta(\tau)$, where $\Delta(\tau)$ is the number of wildtype cells that the mutant has replaced and $\widetilde{N}_{\mathrm{wt}}(\tau) = \frac{4}{3}\pi x_{\mathrm{wt}}^3(\tau)$ is the wildtype population size had there been no mutant. While the mutant is surrounded by the wildtype, we have $\Delta(\tau) = \Delta_1(\tau) = \frac{4}{3}\pi x_{\mathrm{m}}^3(\tau)$. After the mutant breaches the wildtype boundary, $\Delta(\tau)$ is given by the intersection formula of two balls, which we denote $\Delta_2(\tau)$ (see SI Text Section 3 for the explicit formula).

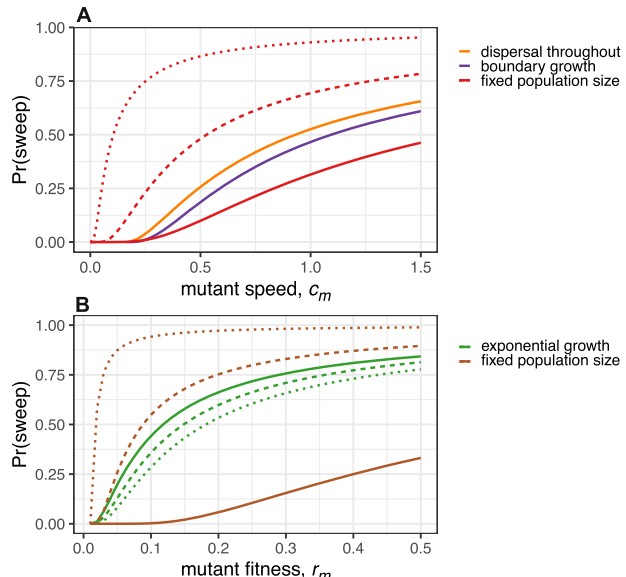

**Fig. 3 | Sweep probabilities for alternative growth models. A** Populations expanding in 3D such that mutants expand within the expanding wildtype population (our main model, orange); proliferation is restricted to the boundary (purple); or the population radius is fixed at $x_0$ (red). For the fixed population size, we set $\mu = 0.23 \times 10^{-5}$ and $x_0 = 20$ (solid curve), 15 (dashed), or 10 (dotted). In the case of an expanding wildtype, we set $c_{\mathrm{wt}} = 0.15$. **B** Well-mixed populations with exponential growth (green) or fixed total population size and logistically growing mutant (brown). We set $r_{\mathrm{wt}} = 0.01$ and $\mu = 10^{-4}$ (solid), $10^{-5}$ (dashed), or $10^{-6}$ (dotted). The fixed population size is $N_0 = 50,000$. Formulas for all curves are summarised in Supplementary Table S2.

Together, we have

$$N_{\mathrm{wt}}(\tau) = (\widetilde{N}_{\mathrm{wt}}(\tau) - \Delta_1(\tau))\mathbf{1}_{[0,\tau_1]}(\tau) + (\widetilde{N}_{\mathrm{wt}}(\tau) - \Delta_2(\tau))\mathbf{1}_{[\tau_1,\tau_2]}(\tau), \tag{6}$$

where $\tau_1$ is the time at which the mutant reaches the wildtype boundary and $\tau_2$ is the time at which the mutant has entirely replaced the wildtype. Noting that

$$\tau_1 c_m = (x - y) + \tau_1 c_{\mathrm{wt}} \Rightarrow \tau_1 = \frac{x - y}{c_m - c_{\mathrm{wt}}} \tag{7}$$

and

$$\tau_2 c_m = (x + y) + \tau_2 c_{\mathrm{wt}} \Rightarrow \tau_2 = \frac{x + y}{c_m - c_{\mathrm{wt}}}, \tag{8}$$

we express $N_{\mathrm{wt}}(\tau)$ in terms of $x$ and $y$ (see SI Text Section 3). We then apply Claim 1 to obtain the conditional sweep probability,

$$\Pr(\text{sweep}|X = x, Y = y) = e^{-\mu \int_0^\infty N_{\mathrm{wt}}(\tau)d\tau}. \tag{9}$$

Although this integral can be solved analytically (SI Text Section 3), the resulting expression is complicated and not very enlightening and we therefore seek simpler approximations. An especially fruitful approach is to assume $y = 0$, so that $f_Y(y|X = x) = \delta(y)$, where $\delta$ is the Dirac delta function. This yields an upper bound on the conditional sweep probability because, due to geometrical symmetry, the time required for a mutant to sweep ($\tau_2$ in eqn. (8)) is minimal when $y = 0$. With this approximation, eqn. (9) simplifies drastically as $\tau_1 = \tau_2 = \frac{x}{c_m - c_{\mathrm{wt}}}$ and we

do not need to integrate over $\Delta_2(\tau)$. The analytic solution is

$$\Pr(\text{sweep}|X = x, Y = 0) = e^{-\left(\frac{x}{\alpha}\right)^4}, \tag{10}$$

where $\alpha = \sqrt[4]{\frac{3(c_m - c_{\mathrm{wt}})^3}{\pi\mu(c_{\mathrm{wt}}^2 - 3c_{\mathrm{wt}}c_m + 3c_m^2)}}$ is a characteristic length. Despite being based on a seemingly crude approximation, eqn. (10) is remarkably close to the exact conditional sweep probability (Fig. 2D). The corresponding approximations in one and two dimensions (SI Text Sections 5 and 6) are likewise simple and useful.

## Unconditional sweep probability

We now have all the necessary ingredients to compute the unconditional sweep probability by marginalising out $X$ and $Y$ from the conditional sweep probability,

$$\Pr(\text{sweep}) = \int_0^\infty \int_0^\infty \Pr(\text{sweep}|X = x, Y = y) f_Y(y|X = x) f_X(x)\, dy\, dx$$
$$= \int_0^\infty \int_0^x e^{-\mu \int_0^\infty N_{\mathrm{wt}}(\tau)d\tau} \frac{3y^2}{x^3} \frac{4x^3 e^{-x^4/\theta^4}}{\theta^4}\, dy\, dx. \tag{11}$$

As before, we use the approximation $y = 0$ to obtain the strikingly simple result

$$\begin{aligned}
\Pr(\text{sweep}) &\leq \int_0^\infty \Pr(\text{sweep}|X = x, Y = 0) f_X(x)\, dx \\
&= \int_0^\infty e^{-\left(\frac{x}{\alpha}\right)^4} \frac{4x^3 e^{-x^4/\theta^4}}{\theta^4}\, dx \\
&= \left(\frac{c_m - c_{\mathrm{wt}}}{c_m}\right)^3.
\end{aligned} \tag{12}$$

This result generalises to analogous one- and two-dimensional models, with the exponent 3 replaced by the respective spatial dimension (SI Text Sections 5 and 6). The approximate expressions are close to the exact solution in one dimension and to numerical evaluations of the integral in two and three dimensions (Fig. 2E). In the one-dimensional case, we can moreover obtain an even better approximation by setting $y^2 = \mathbb{E}[Y^2] = x^2/3$ or (better still[49]) $y^2 = 0.28125x^2$ instead of $y = 0$ and hence shed light on why the upper bound is close to the exact solution (SI Text Section 5). The sweep probability is independent of the mutation rate, not only in these approximations but also in the general case of Eqn. (11). This follows from a more general result (SI Text Section 4). Eqn. (12) also leads to a simple upper bound on the probability of multiple sequential sweeps (SI Text Section 10).

When the expanding wildtype must displace a resident competitor (as in our agent-based simulations, to follow), the sweep probability can be approximated in terms of evolutionary parameters. For example, if we take the speed predictions from the Fisher-Kolmogorov-Petrovsky-Piscounov (FKPP) equation $c_{\mathrm{wt}} = 2\sqrt{Da_{\mathrm{wt}}}$ and $c_m = 2\sqrt{Da_m}$ with $a_{\mathrm{wt}} = r_{\mathrm{wt}} - r_{\mathrm{re}}$ and $a_m = r_m - r_{\mathrm{wt}}$ (see 'Methods') and insert them into eqn. (12) then we obtain

$$\Pr(\text{sweep}) \approx \left(1 - \sqrt{\frac{r_{\mathrm{wt}} - r_{\mathrm{re}}}{r_m - r_{\mathrm{wt}}}}\right)^d, \tag{13}$$

where $r_m$, $r_{\mathrm{wt}}$ and $r_{\mathrm{re}}$ are the proliferation rates of the mutant invader, wildtype invader and resident populations, respectively and $d$ is the spatial dimension.

## Conditional arrival time of the first mutant

In biological systems where it is infeasible to track evolutionary dynamics, selective sweeps must be inferred from subsequent genetic data. For example, we might observe a public mutation in a tumour and ask when this mutation occurred. We can use our model to obtain

the probability distribution of the radius $X$ at the time the first mutant arose, given that we observe a selective sweep, by applying Bayes' theorem,

$$f_X(x|\text{sweep}) = \frac{\text{Pr}(\text{sweep}|X=x)f_X(x)}{\text{Pr}(\text{sweep})}. \quad (14)$$

Using the approximation $y = 0$, we obtain the Weibull distribution

$$f_X(x|\text{sweep}) \approx \frac{4x^3}{\beta^3\theta^4}e^{-\frac{x^4}{\beta^3\theta^4}}, \quad (15)$$

where $\beta = \frac{c_m - c_{wt}}{c_m}$. This approximation and the corresponding results in one and two dimensions are close to the exact solutions (Fig. 2F).

## Sweep probabilities in alternative growth models

To examine the robustness of our findings to variations in the model assumptions, we compare them to results obtained from alternative growth models. Instead of permitting mutants to grow within the wildtype population, one might instead assume that dispersal is restricted to the wildtype population boundary. In this case, a complete sweep is impossible and we instead ask whether the first arising mutant envelopes the wildtype. Interestingly, this envelopment probability is independent of the mutation rate and obtains values close to the sweep probabilities of our main model[36] (Fig. 3A and SI Text Section 7). Although our focus is on expanding populations, we also applied our methods to compute the sweep probability in constant populations (SI Text Section 7), obtaining a result that depends on both mutation rate and population size (Fig. 3), in agreement with prior analyses[50,51].

We have also investigated sweep probabilities in non-spatial models. In the case of exponential growth with no competition, we find that a single mutant is unlikely to become dominant unless its exponential growth rate is several times larger than that of the wildtype and that the sweep probability is relatively insensitive to the mutation rate (Fig. 3B and SI Text Section 7). In the case of constant population size and a logistically growing mutant, the sweep probability is sensitive to both population size and mutation rate (Fig. 3B and SI Text Section 7). For a non-spatial population of constant size, the sweep probability decreases as the mutation rate increases, as in the spatial case. In summary, we find that the main conclusions drawn from our primary model also hold for alternative models of range expansions (exponential or boundary growth) but not for non-growing or tightly bounded populations.

## Validation of our macroscopic model using agent-based simulations

To gauge the robustness of our theoretical predictions to the effects of stochastic growth and discretization, we measure the frequency of selective sweeps in a two-dimensional agent-based model. We suppose that the wildtype population invades a habitat initially occupied by a resident competitor, which is a plausible biological scenario for both invasive species[13] and invasive tumours[26,30]. Our agent-based model has resident, wildtype invader and mutant invader populations with local proliferation rates $r_{re}$, $r_{wt}$ and $r_{m,i}$. Localised competition between wildtype and resident slows the wildtype expansion and creates potential for selective sweeps. For consistency with our macroscopic model, we parametrised the simulations so that all mutations have equal effect ($r_{m,i} = r_m$ for all $i$). We implemented this model using the *demon* agent-based modelling framework[52] within the *warlock* computational workflow[53], which facilitates running large numbers of simulations on a high-performance computing cluster. We have previously applied the same framework to studying cancer evolution[30,53]. Further model details are given in Methods.

The agent-based simulations provide useful insights in addition to the macroscopic model because, although the general setup is the same, they differ in several ways. Space in the simulations is divided into discrete patches; the times between birth and dispersal events are exponentially distributed random variables constituting another source of stochasticity; population boundaries are rough, not smooth; the expansion wave front is typically not sharp and changes shape as the wave progresses; and the mutant will have increased propagation speed when competing with the resident population. Hence, we would not expect perfect agreement between the results of the two models.

Our agent-based model approximately resembles a spatial death-birth Moran process (also known as the stepping stone model)[38,54]. Expansion speeds in the spatial Moran process can in turn be approximated using the FKPP equation[46,47], which predicts that the mutant will expand within the wildtype with constant radial expansion speed $c_m$ dependent on the difference in their local proliferation rates $a_m = r_m - r_{wt}$[42,43,45]. Analogously, we have a constant expansion speed of the wildtype $c_{wt}$ dependent on $a_{wt} = r_{wt} - r_{re}$. To compare the results of our discrete-space simulations to our continuous-space macroscopic model, we measured the propagation speeds of the wildtype within the resident and of the mutant within the wildtype (see 'Methods'). Further investigations of propagation speeds in this model are the subject of a manuscript in preparation.

Given the considerable differences between the models, the probability density functions resulting from the macroscopic and microscopic models are reassuringly consistent. The radius at the time the first surviving mutant arises is slightly lower in the simulations than in our analytical model (Fig. 4A). Similarly, the distribution for the location of the first surviving mutant coincides well except for a small offset of the mean (Fig. 4B). Such offsets are expected due to discretization effects and the fact that, in the simulations, the propagation front needs to be established before the expansion can proceed. The sweep probabilities in simulations are nevertheless very close to our analytical predictions and change very little when varying the mutation rate over orders of magnitude, confirming our prediction that the sweep probability is independent of the mutation rate (Figs. 4C and Supplementary Figs. S1–S4).

## Random fitness effects

Since the effects of mutations are, in reality, not equal but random, we next used our agent-based model to test whether our predictions remain informative when mutation fitness effects are drawn from an exponential distribution, while still assuming that no mutant can accumulate more than one mutation. Because the random fitness effect determines not only a mutant's expansion speed but also its probability of evading stochastic extinction, it is more appropriate to compare results in terms of the mean fitness effect $\tilde{s}$ of the contending mutations that evade stochastic extinction, rather than the mean fitness of all generated mutations[55,56] (see 'Methods'). In terms of this $\tilde{s}$ (Fig. 5C) or an alternative summary statistic (Supplementary Fig. S5; SI Text Section 9), sweep probabilities in random-fitness-effect simulations are close to the predictions obtained under the assumption of fixed effects. Although it is possible for many fitter second or subsequent mutants to arise in the wildtype population and achieve a selective sweep by replacing the wildtype and all less-fit competitor mutants, in our simulations this scenario contributes very little to the total sweep probability.

## Multiplicative fitness effects

Next, we examined the consequences of allowing individuals to acquire multiple mutations with multiplicative fitness effects. The accumulation of beneficial mutations has two opposing effects on the sweep probability. By acquiring further mutations, a mutant lineage can become fitter, expand faster and hence increase its probability of achieving a selective sweep. On the other hand, competitor lineages

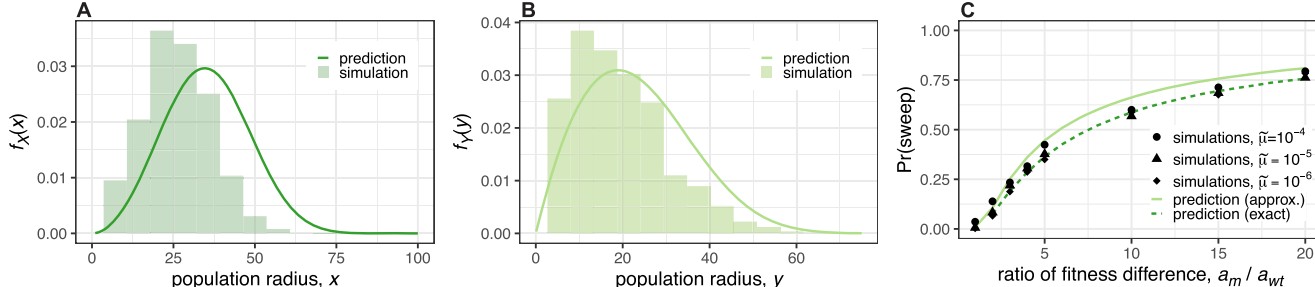

**Fig. 4 | Agent-based simulation results versus macroscopic model predictions.**
**A** Probability density of the wildtype population radius at the time the first surviving mutant arises. **B** Probability density of the distance between the wildtype origin and the location at which the first successful mutant is born. **C** Unconditional selective sweep probability versus the ratio of the proliferation rate differences $a_{wt} = r_{wt} - r_{re}$ and $a_m = r_m - r_{wt}$. Each histogram in (**A**) and (**B**) and each data point in (**C**) is based on 1000 replicates. Except where parameter values are explicitly varied, we set $m = 0.05$, $\tilde{\mu} = 10^{-5}$, $K = 16$, $r_{re} = 0.91$, $r_{wt} = 1$ and $r_m = 1.3$, leading to speeds $c_{wt} \approx 0.15$ and $c_m \approx 0.31$ and mutant survival probability $\rho \approx 0.23$. Results for alternative parameter values and less stringent sweep definitions are shown in Supplementary Figs. S1-S3. The conversion from simulation parameters to macroscopic model parameters is explained in Methods.

can also become fitter and more likely to prevent a sweep. By grouping simulations that have the same distribution of mutation fitness effects, we find that the first factor outweighs the second, so that sweep probabilities are higher in the case of cumulative mutation effects (Fig. 5C). This result makes intuitive sense because the lineage closest to achieving a sweep represents the largest target for further fitness-enhancing mutations. Allowing the accumulation of beneficial mutations also increases the median time at which sweeps finish, in particular when fitness advantages are weak (Fig. 5D and Supplementary Figs. S11–S14). Nevertheless, the conclusion remains that, even when mutations have multiplicative effects, selective sweeps are infrequent unless contending mutations are on average strongly beneficial, in which case sweeps occur early.

When the number of mutations that can accumulate is unlimited, we find that, for every distribution of fitness effects, a single selective sweep is a more common outcome than two sweeps and two sweeps occur more often than three (Fig. 5E). Only rarely do we observe more than three sweeps. This conclusion is robust to the caveat that, for small $s$ values, we likely underestimate the frequency of multiple sweeps because some would complete after we terminate the simulations. By plotting sweep probability against the mean fitness increase over the course of the simulation, we find that sweeps are more often due to a single large-effect mutation than several mutations with small effects (Supplementary Fig. S5).

### Application to cancer

Our findings have especially interesting implications for understanding cancer evolution, in which case we consider a wild-type tumour that evolves while invading a resident population of normal cells. The macroscopic model then has only three parameters: the mutation rate conditional on survival, $\mu$ and the wildtype and mutant propagation speeds, $c_{wt}$ and $c_m$. To estimate $c_{wt}$ for human tumours, we follow a similar procedure to prior studies[36,57]. Consider a tumour that grows to a volume of $V$ between 1 and 10 cm$^3$ in time $T$ between 5 and 20 years. The propagation speed can then be estimated as

$$\tilde{c} = \frac{r}{T} = \frac{\sqrt[3]{\frac{3}{4\pi}V}}{T}, \quad (16)$$

which equates to between 1 and 40 $\mu m$ per day. Given that the diameter of a typical cancer cell is $l \approx 20\,\mu m$[58] and the generation time (cell cycle time) is $\tau_G \approx 4$ days[59], we can switch units to obtain $c = \tilde{c} \times \tau_G / l$, which is between 0.15 and 7 cell diameters per generation. The rate of acquiring advantageous (driver) mutations is usually estimated at around $\tilde{\mu} = 10^{-5}$ per cell per generation[57,60]. For the survival probability $\rho$, we assume values between 0.09 and 0.5, in agreement with inferred values in colorectal tumours[61]. Together, these parameter values lead

to a typical length scale $\theta = \sqrt[4]{\frac{3c_{wt}}{\pi\mu}} \approx 10$ to 50 cells. The expected values of $X$ and $Y$ are then $\mathbb{E}[X] = 9$ to 45 cell diameters and $\mathbb{E}[Y] = 7$ to 35 cell diameters. It follows that a tumour, having acquired sufficient mutations to grow, will likely gain further driver mutations already during early development.

By eqn. (12), if a mutant expands 10% faster within a tumour than the tumour expands into surrounding tissue, then the sweep probability is predicted to be less than $\left(1 - \frac{1}{11}\right)^3 \approx 0.00075$. If the mutant propagates twice as fast as the wildtype, we have Pr(sweep) < 0.13 and in the extreme case where $c_m$ is ten times $c_{wt}$, we have Pr(sweep) < 0.73. In the latter case, the expected population radius when the sweep began, given that a sweep occurred, is approximately 50 cells, corresponding to a population size $N$ of 400,000 cells. The time for the sweep to be completed is then $\tau_2 \approx 40$ generations, at a population size of approximately 800,000 cells. This result is relatively robust because the expected values $\mathbb{E}[X]$, $\mathbb{E}[Y]$ and $\mathbb{E}[X|\text{sweep}]$ are proportional to the characteristic length $\theta$, which varies with the fourth root of $\mu$ and $c_{wt}$. Hence, in three dimensions, changing $\mu$ or $c_{wt}$ by a factor of 10 changes our estimates of the radius and time by only a factor of $10^{1/4} \approx 1.8$ and estimates of the population size by a factor of $10^{3/4} \approx 5.6$. In summary, our macroscopic model (assuming equal mutation effects and no accumulation of mutations) predicts that selective sweeps during a clonal expansion are rare unless mutations are very strongly beneficial, in which case sweeps begin and end early in the expansion. This general conclusion also holds for alternative growth models (Fig. 3).

## Discussion

Here, we have used mathematical modelling and analysis to determine the expected frequency of selective sweeps. We find that this frequency is generally expected to be low, even for mutations with a strong selective advantage. Moreover, when the wildtype and mutant radial expansion speeds are constant, the sweep probability can be expressed solely in terms of those speeds, which can in turn be related, through the FKPP equation or other standard models, to the selection coefficient, dispersal rates and other basic parameters. Our analytical predictions remain informative even when mutation fitness effects are random and multiplicative.

Why is the sweep probability independent of the mutation rate? An intuitive explanation is that if the mutation rate is higher, then on the one hand, the first advantageous mutation is likely to arise in a smaller population—meaning that it has less distance to travel to achieve a sweep— but, on the other hand, competing mutations will also tend to arise sooner than in the case of a lower mutation rate. These two effects exactly cancel out under the assumption of constant radial growth speeds. In alternative growth models, the two effects are unequal, resulting in either a positive or a negative effect of mutation rate on sweep probability (SI Text Section 8).

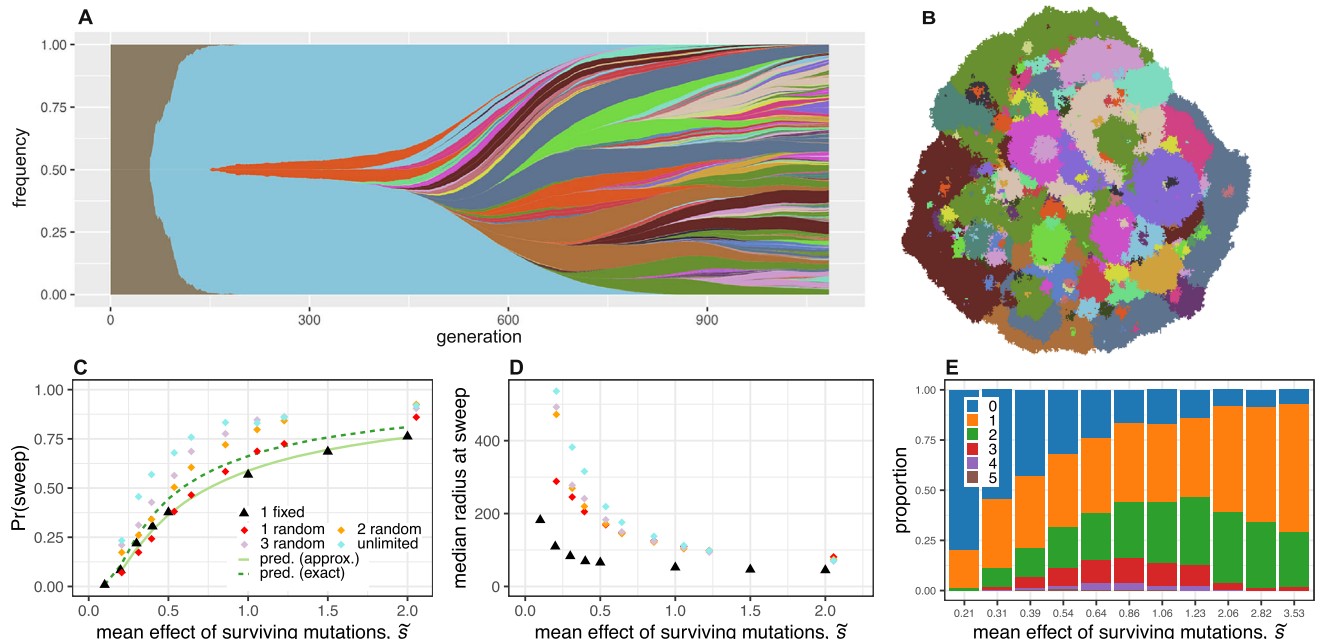

**Fig. 5 | Results of models with random and cumulative mutation effects.**
**A** Mutant frequencies plotted against time for a two-dimensional agent-based simulation in which mutations have multiplicative random effects on fitness and the accumulation of mutations is unrestricted. Distinct mutants are represented by different colours, and descendants are shown emerging from within their parent population. In this case, the first-arising mutant (pale blue) achieves a sweep by replacing the wildtype (grey-brown), before itself being replaced by numerous descendants. **B** Spatial arrangement of mutants at the end of the same simulation. Each pixel corresponds to a well-mixed deme containing approximately $K = 16$ individuals, and is coloured according to the most abundant mutant using the same colour scheme as (**A**). **C** Unconditional selective sweep probability for the two-dimensional model versus mean fitness effect of surviving mutations (those that evaded stochastic extinction), for simulations in which individuals can acquire a maximum of 1, 2, 3 or an unlimited number of mutations. A sweep is counted irrespective of whether it was achieved by the first, the second, or a subsequent arising mutation. Outcomes have been adjusted to account for the estimated proportion of sweeps that would have finished after the simulation stop time (see 'Methods'). **D** Median population radius at the time of a successful sweep versus mean fitness effect of surviving mutations (based on fitting gamma distributions to the simulation results; see 'Methods'). **E** Multiple sweep frequencies in simulations in which individuals can acquire an unlimited number of mutations. Parameter values for all plots are $m = 0.05$, $\mu = 10^{-5}$, $K = 16$, $r_{re} = 0.91$ and $r_{wt} = 1$.

We make three arguments to justify our focus on this particular model of range expansions. First, our model corresponds to the continuum approximation of standard mathematical models of range expansions and spatial population genetics: the spatial Moran process (or stepping stone model[54]) and the biased voter model (which is equivalent to an Eden growth model extended to allow local dispersal and competition throughout the population)[45]. These models are well understood, intuitive and easy to parameterise. Second, because our model is relatively permissive to selective sweeps, it provides useful upper bounds for selective sweep probabilities in more complex scenarios, as further explained below. Our third justification is that, in much the same way as the Moran and Wright-Fisher processes are the most useful, tractable models of evolution in constant-sized, non-spatial populations, so the constant-radial-speed model yields the clearest results for range expansions. Haldane's famous rule of thumb is that the fixation probability of a weakly beneficial allele in a large, non-spatial population of constant size is approximately proportional to its relative advantage in terms of proliferation rate[62]. Here, we have obtained the comparably simple result that the probability of a strongly beneficial allele achieving a selective sweep in an expanding population is approximately equal to its relative advantage in terms of radial expansion speed, raised to the power of the spatial dimension. For instance, if the radial expansion speed at which a mutant spreads within the wildtype population is twice the speed at which the wildtype expands, then the probability of this mutant achieving a selective sweep can be approximated simply as $(1-1/2)^2 = 1/4$ in two dimensions and $1/8$ in three dimensions.

Some alternative models have been considered previously. Antal and colleagues[36] used a macroscopic model similar to ours to investigate the case in which mutations arise only at the boundary of a range expansion. Given that selective sweeps are then impossible, the interesting outcome is when the mutant envelops the wildtype. Ralph & Coop[50] and Martens and colleagues[35,51] instead considered constant-sized populations and found that selective sweeps are likely only if the population width is much smaller than a characteristic length scale, which depends on the mutation rate, the dispersal rate, the effective local population density and the strength of selection. In SI Text Section 7 we compare our results to the findings of these and other prior studies, including a well-known result of Gerrish and Lenski for non-spatial populations[55]. The general conclusions are that selective sweeps are predicted to be rare except in small populations and that our model provides a useful upper bound on the sweep probability in range expansions.

A selective sweep can occur only if the rate of spread of an advantageous mutation exceeds the expansion speed of the wildtype population. This scenario is plausible, for example, in a biofilm growing slowly under antibiotic stress, in which case our findings predict the evolutionary dynamics of antibiotic resistance. Our results apply equally to an invasive species that is still adapting to its new conditions and whose range expansion is slowed by the need to modify its environment (niche construction) or by interspecific interactions[13]. If the invader must outcompete a resident, then the sweep probability can be approximated via the FKPP solution in terms of proliferation rates as in eqn. (13). The dimensional exponent in this result implies that selective sweeps are most likely to occur in species invading essentially linear habitats, such as coastlines.

Parametrised for human solid tumours, our macroscopic model indicates that selective sweeps during clonal expansions are restricted to strong driver mutations during the early stages of expansion. Late-

arising drivers can become locally abundant but are unlikely to become fixed. There are several reasons to expect this general conclusion to hold even when driver mutation effects are random and multiplicative. First, an extension of our mathematical analysis suggests that the increase in sweep probability due to the accumulation of beneficial mutations is not much larger in three dimensions than in our two-dimensional simulations (Supplementary Fig. S17). Second, selective sweeps will be less frequent when the wildtype radial expansion speed increases over time (as is typical in the middle stage of tumour progression) or when mutant expansion is restricted to the tumour boundary (which may be the case in some tumours[30,33,63]). Third, whereas our models assume a homogeneous microenvironment, tumours typically contain regions of hypoxia, necrosis and connective tissue that slow or prevent cell dispersal and hence impede selective sweeps. Fourth, due to microenvironmental heterogeneity and niche construction, fitness landscapes are likely to vary such that a mutation that is beneficial in one tumour region may be neutral or deleterious in another[64].

Our prediction that selective sweeps are rare during later tumour expansion, even if driver mutations continue to accumulate, is consistent with recent findings of the Pan-Cancer Analysis of Whole Genomes consortium. Analysing 2658 human tumour genomes, the consortium detected on average four clonal driver mutations per tumour[21,22], while also finding that more than 95% of tumours exhibit at least one subclonal expansion[65]. Although only 11% of subclonal expansions could be explained by single-nucleotide variants that are known cancer drivers, the consortium found evidence of positive selection across subclones and cancer types. They therefore concluded that tumours may harbour additional subclonal expansions (corresponding to incomplete sweeps) driven by copy number aberrations, structural variants, epigenetic alterations, or single-nucleotide variants that have yet to be identified as drivers[65].

How then do tumours acquire multiple clonal driver mutations? Driver mutations can sometimes be classified into early and late events by leveraging point mutations on top of copy-number gains[22,23]. However, because a selective sweep resets the common ancestor, it has proven difficult to determine whether clonal drivers are mostly acquired before or during tumour expansion. According to our models, it is highly unlikely that more than two selective sweeps will occur during a tumour's final growth phase. Multiple clonal driver mutations are therefore better explained by multi-stage models of tumour initiation and progression (reviewed in ref. 66), in which growth repeatedly stalls due to constraints such as hypoxia, immune control and physical barriers. Driver mutations enable subclones to escape these constraints and invade new territory, each time purging genetic diversity so that the final, prolonged expansion originates from a single highly transformed cell. This episodic model is conventional[39,67] and has been particularly well characterised in recent studies of colorectal cancer[68,69] and breast cancer[70]. Our results suggest that once a tumour has entered its final growth phase and is more than a cubic millimetre in volume, even extremely strong drivers are highly unlikely to become clonal and will instead contribute to genetic heterogeneity and possibly to parallel evolution[71].

The mathematical approach we have developed here can potentially be extended to investigate the dynamics of mutant population sizes, to understand better how intratumour heterogeneity relates to cancer treatment outcomes[17,26] and to develop more effective cancer treatment strategies[72,73]. To obtain more precise predictions rather than upper bounds, it will be important to examine how microenvironmental heterogeneity, immune responses and phenotypic plasticity inhibit clonal expansions in each cancer type. Our predictions also motivate further investigation of selective sweeps in biofilms and other experimental systems[2].

## Methods

### Numerical integration
We performed numerical integration using the MATLAB function 'trapz'. For values of $x$ close to 0 and beyond $2\theta$, $f_X(x)$ is small and its contribution to the integrals is negligible. Hence, to avoid numerical errors without compromising precision, we set the lower and upper bounds of integration to $0.001\theta$ and $3\theta$ when integrating over $f_X(x)$. Similarly, we set $0.001\theta$ as the lower bound of the integration over $y$. We set interval widths to $0.001\theta$ for $x \leq 0.5$ and $0.01\theta$ for $x > 0.5$.

### Agent-based simulations
We ran agent-based simulations using the Warlock automated computational workflow for the demon modelling framework[30,53]. Individuals in this agent-based model are subdivided into well-mixed demes on a regular two-dimensional grid. The demes have identical carrying capacities, $K$ and are initially filled with residents, except that a single wildtype invader is introduced at the centre of the grid. At each time step, an individual is chosen at random, with probability weighted by fitness, to be replaced by two offspring. Each offspring then either migrates, with probability $m$, to a neighbouring deme in a randomly chosen direction or remains in its parent deme. Local density dependence is implemented by imposing a very high death rate whenever a deme is above the carrying capacity. Mutation is coupled to wildtype reproduction. Motivated by an invasive cancer cell population spreading in non-invasive healthy tissue, we do not permit resident individuals to disperse. We can account for this asymmetry by adapting the conversion from fitness advantage into expansion speeds (to be published in a later study). Further model details have been published previously[30,53]. All simulations were performed using City, St George's, University of London's Hyperion cluster. Example graphical outputs of the simulations, plotted using the ggmuller R package[74], are included in Supplementary Figs. S7 and S8.

For all simulations, we set the deme size to $K = 16$ and the migration probability $m = 0.05$. For such a small $m$, the probability of surviving drift can be estimated as the probability of becoming locally fixed in one deme. Since the within-deme dynamics approximate a Moran process, this probability is $\rho = \frac{1 - r_{wt}/r_m}{1 - (r_{wt}/r_m)^K}$, where $r_{wt}$ and $r_m$ are the proliferation rates[75]. We measured propagation speeds by simulating the expansion of a wildtype into a large resident population, or of a mutant into a large wildtype population, in the absence of mutation and then applying linear regression to the growth curve of the effective radius, defined as the square root of the population size divided by $\pi$.

For each parameter set, we took the mean speed from ten replicates, for which the standard deviation was consistently below 2% of the mean. We set the proliferation rates to $r_{re} = 0.91$ and $r_{wt} = 1.0$ for the resident and wildtype, respectively, leading to $a_{wt} = r_{wt} - r_{re} = 0.09$. The measured wildtype speed was then $c_{wt} \approx 0.15$. In simulations with fixed mutation effects, we varied the mutant proliferation rate $r_m$ from 1.1 to 3.0, so that $a_m = r_m - r_{wt}$ ranged from 0.1 to 2.0. The measured mutant speeds are presented in Supplementary Table S3.

To implement random mutation effects, we multiplied the fitness effect of each mutation by a factor $X$ drawn from an exponential distribution with mean 1. To prevent birth rates from becoming implausibly large, we assumed diminishing-returns epistasis associated with maximum birth rate $M = 10$. The birth rate after $i$ mutations was then

$$r_{m,i} = \min\left\{ r_{m,i-1}\left(1 + sX\left(1 - \frac{r_{m,i-1}}{M}\right)\right), M \right\}, \tag{17}$$

where parameter $s > 0$ determines the mean mutation effect.

## Computation of sweep probabilities in simulations with fixed mutation effects

We ran a batch of 1000 simulations for each combination of mutation rate $\widetilde{\mu} \in \{10^{-4}, 10^{-5}, 10^{-6}\}$ and mutant proliferation rate $r_m \in \{1.1, 1.2, 1.3, 1.4, 1.5, 2.0, 2.5, 3.0\}$. Resident and wildtype proliferation rates were in all cases $r_{re} = 0.91$ and $r_{wt} = 1.0$. Simulations were stopped at a population size of 1,000,000 or at 2000 generations (whichever was reached first). We called a sweep when all individuals at the end of the simulation shared a mutation.

To investigate the extent to which we failed to detect selective sweeps that would have completed after the simulation stop time, we examined the population sizes when the detected sweeps finished. For every set of parameter values for which sweeps were abundant enough to analyse, the population radius at the time of sweep completion exhibited a bell-shaped distribution that decayed steeply before the maximum possible radius $\sqrt{10^6/\pi}$, indicating that we detected the vast majority of sweeps. In the remaining cases ($\widetilde{\mu} = 10^{-5}$ and $r_m = 1.1$; $\widetilde{\mu} = 10^{-6}$ and $r_m = 1.1$ or 1.2), sweeps were sufficiently rare that even relatively severe undercounting would not affect our conclusions.

## Computation of sweep probabilities in simulations with random mutation effects

For each variant of the random-mutation-effects model (permitting the accumulation of up to one, two, three or unlimited mutations), we ran eleven batches of 1000 simulations, with fitness effects calculated according to eqn. (17) and $s \in \{0.05, 0.075, 0.1, 0.15, 0.2, 0.3, 0.4, 0.5, 1, 1.5, 2\}$. As before, simulations were stopped when they reached a population size of 1,000,000 or 2000 generations.

The establishment probability $\rho$ is lower for weakly beneficial than for highly beneficial mutations, which implies that the effective mutation rate $\mu = \rho\widetilde{\mu}$ is lower in the former case. For example, in a Moran process with $K = 16$, nearly 94% of nearly-neutral mutations succumb to stochastic extinction before they can contribute to the macroscopic evolutionary dynamics, compared to only 50% of mutations that confer fitness effects $s = 1$. To account for this difference in effective mutation rates, we calculated, in the at-most-one-mutation model, the mean effect $\widetilde{s}$ of mutations that reached an abundance of at least 10 individuals (see Supplementary Fig. S16 for the distributions). We then compared the sweep probability for each random-effects model with that of the equal-effects model with $a_m = \widetilde{s}$. As before, we transformed $a_m$ into $c_m$ according to Supplementary Table S3.

To investigate whether our finite-time simulations omit sweeps that would occur if simulations were run for longer times, we investigated the population radii at sweep completion. Population size rather than number of generations was the dominant stopping criterion except when $s = 0.05$ in all models and $s = 0.075$ in the at-most-one-mutation model (Supplementary Fig. S9). In all cases, the radius distributions are bell-shaped and indicate that, for all but the smallest fitness effects, most selective sweeps finished long before the simulation stop time (Supplementary Figs. S11–S14). To estimate the extent of undercounting, we fitted right-truncated gamma distributions to the radius distributions, with the truncation at $\sqrt{1,000,000/\pi} \approx 564$ corresponding to the population size at which simulations were stopped. Using a maximum likelihood approach, we then estimated the scale and shape parameters of each gamma distribution. By evaluating the probability in the tail of the non-truncated gamma distribution beyond the truncation point, we estimated the fraction of sweeps omitted in each batch of simulations and adjusted the sweep probabilities in Fig. 5C accordingly. The adjustments are negligible except in the case of small fitness effects, for which sweep probabilities are in any case low. In tests using artificially truncated distributions from simulations with large $s$ values (Supplementary Fig. S15), we find that this method typically overestimates the proportion of missing sweeps. The adjusted values can therefore be regarded as upper bounds.

## Data availability

Configuration files and simulated data can be downloaded from https://doi.org/10.5281/zenodo.10775383.

## Code availability

Supporting Mathematica scripts and R code to generate the figures can be found at https://doi.org/10.5281/zenodo.18246214. Simulations were run using the workflow *warlock*, which is based on the *demon* simulation framework. The code can be found at https://doi.org/10.5281/zenodo.7435093 and https://github.com/robjohnnoble/demon_model respectively.

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

## Acknowledgements

We thank Jonas Demeulemeester and Maxime Tarabichi for guidance on the interpretation of recent cancer sequencing studies, and Guillaume Martin for helpful discussions about mutation fitness effect distributions. We are grateful for the use of City St George's Hyperion cluster to run the many simulations integral to this study. A.S. was supported by the European Union's Horizon 2020 research and innovation programme under the Marie Składowska-Curie EvoGamesPlus grant agreement no. 955708, and by UK Research and Innovation (UKRI) under grant no. MR/V02342X/1. M.B. was supported by an award from the City of St George's, University of London Research Pump-priming Fund. R.N. was supported by the National Cancer Institute of the National Institutes of Health under Award Number U54CA217376. The content is solely the responsibility of the authors and does not necessarily represent the official views of the National Institutes of Health.

## Author contributions

R.N. conceived the research question and supervised the project. R.N. and A.S. designed the research. A.S. and R.K. carried out the mathematical analysis. M.B. and K.B. performed agent-based simulations. A.S. and K.B. analysed simulation results. All authors wrote and approved the manuscript.

## Competing interests

The authors declare no competing interests.
