## [Transparent Peer Review file · Nature Communications]

Selective sweep probabilities in spatially expanding populations

Corresponding Author: Dr Robert Noble

Version 0:

Reviewer comments:

Reviewer #1

(Remarks to the Author)

Stein, Kizhuttill, Back, and Noble present a mathematical model for studying how an advantageous mutation spreads in a wildtype population and dominates it. The authors are interested in the sweep probability, which is the probability that the mutant and its descendants replace the wildtype population. The main scenario consists of wild type and mutant populations that expand radially in space. The authors derived the sweep probability and showed that it is independent of the mutation rate. Next, they studied a three-type agent-based model for verification of the theory. In addition, several other spatial and non-spatial scenarios are considered for comparison with the main scenario. Lastly, the authors applied the theory to cancer evolution.

The main text is well organized, and the mathematical derivations are easy to follow. The observation that the sweep probability is independent of the mutation rate is enlightening.

Here are some other comments for the main text:

1. Figure 2: Why is the shape of 1-d Y density so different from the 2-d and 3-d Y density? It seems that the formulas have a common form of the incomplete Gamma function.
2. Line 157,
$$(8): \tau_1$$
 before the double arrow should be τ_2 .
3. Line 184: It would be nice if the authors give the full name of FKPP here. (There is one in line 223 but it might be better to put it here.)
4. Line 184,
$$(13):$$
 It's hard to find the derivation of this approximation in the main text and the SI. Can the authors point it out explicitly here?
5. Figure 3, panel B: The colors are different from panel (A) so there should be a legend here. In addition, could the author provide the number of simulations for generating these plots? It seems that the bin width of the histogram in (A) and (B) are different.
6. Line 262: The location of the parenthesis is incorrect.
7. Line 306: The discussion on the conditions that the sweep probability is independent of the mutation rate is great. I wonder if there are real biological cases in which these conditions are nearly satisfied and we can observe the independence. In addition, do the authors have an intuition on the approximation of locating the first mutant at the center? I think this may be due to the geometric symmetry of the expansion as well as the assumption that the mutant arises uniformly among the population.

There are some typos in the Supplementary:

1. Line 14: The dummy variable of in the integral should not be t .
2. Line 28,
$$[1]:$$
 Parentheses is missing.
3. Line 150: There is a missing reference.
4. Line 154,
$$[32]:$$
 The display formula should be corrected.
5. Line 159,
$$[34]:$$
 The 'wt' in the last line should be a subscript.

(Remarks on code availability)

Reviewer #2

(Remarks to the Author)

In this paper, Stein et al derive mathematical expressions for selective sweep probabilities in simple models of expanding populations. The authors are motivated by tumors, which, through recent multi-site and single cell sequencing efforts, have been revealed to be highly genomically heterogeneous. The authors place this at odds with older models from Vogelstein and others which hypothesize that tumors acquire driver mutations through sequential selective sweeps, which would reduce heterogeneity.

The authors' central claim is that range expansions restrict selective sweeps to small tumors (of the order tens of cells), which they show by analyzing a very simplified model of range expansion. In essence, they model a linearly expanding population (WT) as an initial tumor clone. They allow for a second driver (MT) to emerge at a prescribed mutation rate on this expanding WT background. In general, it is assumed the expansion speed of MT > expansion speed of WT. They define a fixation event as a case where MT completely displaces WT before another mutation arises on the WT background. This gives simple expressions for the fixation probability, since the problem, as laid out, can be cast in purely geometrical terms.

Given this relatively simple setup, the mathematical analysis looks sound to me. And I very much appreciate the attempt to place the model in context by estimating parameters in the "Application to cancer" section.

However, I have broad concerns about the conceptual foundation of the model, i.e. simple models are good, but I am not convinced that this model is sufficiently complex to capture some of the more salient aspects of cancer evolution.

Major concern:

The most serious concern is about the motivation of the model itself -- is this the right model to study if cancer is the point of comparison? It is not clear to me that selective sweeps of the sort described are a relevant scenario in cancer. Rather, selective sweeps function as a sort of straw man in the manuscript. The model as laid out describes a highly restricted scenario, where only mutations on the WT background are accounted for. However, if the MT population is expanding faster or once the MT is larger than the WT (if mutations are per birth or per capita, respectively), it is the MT that is likely to get more subsequent beneficial mutations. These additional drivers on the MT background could abrogate the observed effect of reduced fixation of the MT allele (since drivers emerging on the WT background would have to catch up to carriers of the MT allele that have acquired additional beneficial mutations, assuming a fixed fitness effect size).

This is one relatively small tweak that can have a drastic impact on the conclusions that are drawn from the model. The authors themselves mention in the text and describe in the SI that the key result that the sweep probability is independent of the mutation rate is highly dependent on the details of the growth process. In fact, 'either a positive or a negative effect of mutation rate on sweep probability' can be observed depending on the underlying growth dynamics. Cancers typically grow in highly heterogeneous environments, and exhibit different growth properties depending on the cell type of origin or metastatic site -- so this is likely a relevant perturbation to the model that can drastically change the results.

Indeed, if we look to the literature, it has also been shown that, in spatially structured models parametrized by cancer contexts, local measures of heterogeneity are limited due to cell turnover (cf, Waclaw, et al. 'A spatial model predicts that dispersal and cell turnover limit intratumour heterogeneity'). This is in stark contrast to what the authors conclude -- that spatial structure increases heterogeneity. My understanding is that this is due in part to the fact that the present authors primarily focus on the sweep probability (a global measure of loss of heterogeneity). However, in a spatial context, spatially continuous measures of heterogeneity might be more appropriate to use and to compare, at least over the relevant observation timescales.

Taken collectively, these issues indicate to me that the results are of somewhat restricted applicability, and are highly dependent on the particular, relatively simple formulation of the model.

Minor concern:

Please describe the "warlock" code in more detail, perhaps illustrating what its output looks like -- the manuscript would greatly benefit from having this be described in a self-contained way, even if the software was published elsewhere.

(Remarks on code availability)

The code is contained in a rather large compressed file (>4GB) -- it would be much more preferable to store the code relevant to the paper (which is presumably somewhat simple, and significantly smaller than 4GB), in a Github repository.

Version 1:

Reviewer comments:

Reviewer #1

(Remarks to the Author)

The authors have addressed all my comments, and I recommend the manuscript for publication.

(Remarks on code availability)

The GitHub and Zenodo repositories are well structured and contain Readme files to guide readers. I appreciate that some integrations are verified and documented using Mathematica. Reproducing agent-based simulations is not easy, as it requires generating and analyzing data using the demon package from other publications.

Reviewer #2

(Remarks to the Author)

Stein et al have amended their manuscript in two primary ways, adding more simulation results and expanding introductory and discussion points. I appreciate that effort has been put in to address my points of concern. However, the presentation of some of the additional results (extending the simulation model to random and accumulative fitness effects) come across as confusing, and potentially incorrect. Moreover, the relationship of these new simulation results to the analytical model is somewhat tenuous. Finally, the key predictions of the model wrt its relationship to cancer data seem to be demonstrably inconsistent with observations described in the literature.

Major points:

I will first address the new individual-based simulations (those pertaining to Fig 4 in the revised text). There are two new simulations that are introduced:

1. Simulations in which random effect size mutations are introduced on the WT background, but further mutations cannot accumulate.
2. Simulations in which mutations can continue to accumulate "multiplicatively."

The simulations in which random effect single mutations are introduced are problematic conceptually. The study aims to describe sweep probabilities, which are typically infinite time quantities, but individual-based models can only be run for finite time (presumably 1000 generations in this study, according to figure 4A, though this is never stated). The precise time to see a sweep (and therefore, the appropriate time to run a simulation) will sensitively depend on several underlying parameters like mutation rate, the mode of overall population growth (fixed, boundary-drive, exponential), and the distribution of fitness effects. Therefore, it is not clear that the "macroscopic" model and analytical results derived from it are the appropriate point of comparison for finite time data generated by individual-based simulations in the case where mutation effect sizes vary. Indeed, if the simulations were run for longer, we would eventually see an atypically large effect size beneficial mutation arise and eventually sweep in every simulation by chance (assuming an exponential distribution of effect sizes as the authors state).

Moreover, the "anomaly" at low fitness observed in these simulations and described in Figure 4C is confusing, and I'm not sure the explanation provided in the text is satisfactory. Specifically, what is even being measured in Figure 4C? The probability of the first mutation sweeping or the probability of any mutation sweeping? What is the x-axis precisely? Figure 4D is more clear wrt the x-axis, but it shows that the sweep probability increases as more mutations are allowed to accumulate, which is a more realistic scenario than the analytical model where mutations do not accumulate.

For the simulations that allow for multiplicative accumulation of mutations, the problems are potentially more severe. According to the process by which the proliferation rate of mutations changes (Eq. 17), for some of the values stated (for instance, $s=0.1$) the proliferation rate decreases on average (this can be seen by simulating the process described by Eq. 17). Since further mutations are relatively deleterious on average in this model, it is perhaps not so surprising that the fixation of mutant lineages is somewhat limited. Moreover, simulations with different fitness scales s are apparently pooled to generate Figs 4C-E and may have contributed to the "anomaly" described in the text and above.

Finally, I will highlight a key prediction that the authors bring up, that is seemingly untrue. The statement "we predict... detectable tumours will almost always contain multiple competing clones with different driver mutations" (line 360) is central to the message of the paper (and I appreciate that the authors were bold enough to go out on a limb and say this). Unfortunately, it is likely false. The authors cite a convenient figure from Dentre et al. (that 95% of tumors in PCAWG exhibit subclonal expansion, line 429) but neglect the key observation from that same paper that only 11% of subclones carry a driver. This suggests that subclonal dynamics are driven by considerations other than purely fitness on a fixed landscape per se (for instance, there could perhaps be 'ecological' effects or a deformed fitness landscape). In an ecological scenario, for instance, sweeps would be prevented by more "trivial" mechanisms like disjoint niche preferences. Beyond that, it is clear that there is evidence for "multi-hit" origins of certain cancers, which involve the acquisition of multiple clonal mutations and therefore imply selective sweeps. If the authors were to provide some hard evidence from cancer data to back up some of the key predictions of the model, I would love to see it.

Minor points:

I appreciate that a version of the code has been uploaded to Github. It appears that much of the underlying simulation code resides in a single >4000 line c++ file (https://github.com/robjohnnoble/demon_model/blob/master/src/demon.cpp). Considering that previous versions of this code have ostensibly been reviewed elsewhere, reviewing this is beyond my scope. However, this file is somewhat difficult to navigate and could stand to be more thoroughly commented or broken up into more easily parsed chunks. Moreover, I was not able to get it to run.

(Remarks on code availability)
I was not able to get it to run.

Reviewer #3

(Remarks to the Author)

(Remarks on code availability)

Version 2:

Reviewer comments:

Reviewer #4

(Remarks to the Author)

As I have been brought in relatively late in the review process, I focused my attention on independently assessing how the previous reviewers' comments have been addressed. My assessment is that the authors have carefully resolved the confusion that arose when introducing new results from agent based simulations. They have amended and clarified some potentially problematic statements that were pointed out by the previous reviewers and addressed the concern about the apparent disagreement between some cancer data and their results, specifically in regards to the reference by Dentro et al. In conclusion, I think all the points carefully raised by the previous reviewers have been fully addressed and the manuscript is ready for publication.

The available code has also been carefully amended according to the reviewers' suggestions.

(Remarks on code availability)

The repository appears well structured and easy to navigate. The comments from previous reviewers' have been carefully addressed.

Response to the reviewers' comments

Reviewer #1 (Remarks to the Author):

Stein, Kizhuttill, Back, and Noble present a mathematical model for studying how an advantageous mutation spreads in a wildtype population and dominates it. The authors are interested in the sweep probability, which is the probability that the mutant and its descendants replace the wildtype population. The main scenario consists of wild type and mutant populations that expand radially in space. The authors derived the sweep probability and showed that it is independent of the mutation rate. Next, they studied a three-type agent-based model for verification of the theory. In addition, several other spatial and non-spatial scenarios are considered for comparison with the main scenario. Lastly, the authors applied the theory to cancer evolution.

The main text is well organized, and the mathematical derivations are easy to follow. The observation that the sweep probability is independent of the mutation rate is enlightening.

We are glad that the reviewer found our manuscript to be clear and enlightening.

Here are some other comments for the main text:

1. Figure 2: Why is the shape of 1-d Y density so different from the 2-d and 3-d Y density? It seems that the formulas have a common form of the incomplete Gamma function.

The different shapes can be understood from a mathematical perspective as follows: By definition, the incomplete gamma function is a monotonically decreasing function of $y > 0$. This explains the monotone behaviour in the 1-d case, for which the probability density function (pdf) is simply proportional to the incomplete gamma function (Table S1). In the other cases, the pdf also contains a factor of y (for the 2-d model) or y^2 (3-d). A Taylor expansion shows that the factor of y or y^2 dominates when y is small, so that the pdf is zero when $y=0$ and increases approximately linearly (2-d) or quadratically (3-d) when y is small. When y is large, the incomplete gamma function dominates and the curve decreases faster than exponentially.

2. Line 157, display (8): τ_1 before the double arrow should be τ_2 .

We corrected the typo.

3. Line 184: It would be nice if the authors give the full name of FKPP here. (There is one in line 223 but it might be better to put it here.)

We now do so.

4. Line 184, display (13): It's hard to find the derivation of this approximation in the main text and the SI. Can the authors point it out explicitly here?

We have added the necessary details (lines 195-198).

5. Figure 3, panel B: The colors are different from panel (A) so there should be a legend here. In addition, could the author provide the number of simulations for generating these plots? It seems that the bin width of the histogram in (A) and (B) are different.

We added a legend to panel B and made the bin widths identical. The number of simulation replicates was already stated in the caption (we have put it in red text for ease of reference).

6. Line 262: The location of the parenthesis is incorrect.

We corrected the typo.

7. Line 306: The discussion on the conditions that the sweep probability is independent of the mutation rate is great. I wonder if there are real biological cases in which these conditions are nearly satisfied and we can observe the independence. In addition, do the authors have an intuition on the approximation of locating the first mutant at the center? I think this may be due to the geometric symmetry of the expansion as well as the assumption that the mutant arises uniformly among the population.

Despite searching, we have been unable to find any empirical study of how sweep probabilities in range expansions vary with mutation rate. Detecting selective sweeps in natural systems is generally difficult. One possibility would be to design lab experiments like those of Hallatschek and colleagues using fluorescently labelled *E. coli* (Hallatschek *et al.* *PNAS*, 2007), as we now mention in our Discussion (lines 459-460).

We have added an explanation of why the upper bound on the sweep frequency, obtained by setting $y = 0$, is a good approximation to the exact result in the one-dimensional case. Linked to this explanation, we provide new, better approximations in the one-dimensional case (main text lines 185-188; supplementary text lines 119-128).

There are some typos in the Supplementary:

1. Line 14: The dummy variable of in the integral should not be t .
2. Line 28, display [1]: Parentheses is missing.
3. Line 150: There is a missing reference.
4. Line 154, display [32]: The display formula should be corrected.
5. Line 159, display [34]: The 'wt' in the last line should be a subscript.

We have corrected the typos. Thank you for taking the time to check our manuscript so thoroughly.

Reviewer #2 (Remarks to the Author):

In this paper, Stein et al derive mathematical expressions for selective sweep probabilities in simple models of expanding populations. The authors are motivated

by tumors, which, through recent multi-site and single cell sequencing efforts, have been revealed to be highly genomically heterogeneous. The authors place this at odds with older models from Vogelstein and others which hypothesize that tumors acquire driver mutations through sequential selective sweeps, which would reduce heterogeneity.

The authors' central claim is that range expansions restrict selective sweeps to small tumors (of the order tens of cells), which they show by analyzing a very simplified model of range expansion. In essence, they model a linearly expanding population (WT) as an initial tumor clone. They allow for a second driver (MT) to emerge at a prescribed mutation rate on this expanding WT background. In general, it is assumed the expansion speed of MT > expansion speed of WT. They define a fixation event as a case where MT completely displaces WT before another mutation arises on the WT background. This gives simple expressions for the fixation probability, since the problem, as laid out, can be cast in purely geometrical terms.

Given this relatively simple setup, the mathematical analysis looks sound to me. And I very much appreciate the attempt to place the model in context by estimating parameters in the "Application to cancer" section.

However, I have broad concerns about the conceptual foundation of the model, i.e. simple models are good, but I am not convinced that this model is sufficiently complex to capture some of the more salient aspects of cancer evolution.

We are grateful to the reviewer for raising these concerns, which we have addressed by running extensive new simulations and analyses. We believe that the new results substantially strengthen our manuscript without changing its main conclusions.

Major concern:

The most serious concern is about the motivation of the model itself -- is this the right model to study if cancer is the point of comparison? It is not clear to me that selective sweeps of the sort described are a relevant scenario in cancer. Rather, selective sweeps function as a sort of straw man in the manuscript.

We have rewritten parts of the Introduction (lines 48-69) and Discussion (lines 425-436 and 451-455) and added more references to clarify how and why our model is relevant to cancer.

The model as laid out describes a highly restricted scenario, where only mutations on the WT background are accounted for. However, if the MT population is expanding faster or once the MT is larger than the WT (if mutations are per birth or per capita, respectively), it is the MT that is likely to get more subsequent beneficial mutations. These additional drivers on the MT background could abrogate the observed effect of reduced fixation of the MT allele (since drivers emerging on the WT background would have to catch up to carriers of the MT allele that have acquired additional beneficial mutations, assuming a fixed fitness effect size).

We agree that it is important to examine this scenario. We now report extensive new simulations and analyses of models that permit mutants to accumulate beneficial

mutations with random, multiplicative fitness effects (main text lines 258-309 and 500-508; supplementary text Sections 9 and 10; Figures 4 and S5). We conclude from these new results that, “consistent with the predictions derived from our macroscopic model, a selective sweep is highly probable only if a mutant with a large fitness advantage arises in the very early stages of a range expansion”.

This is one relatively small tweak that can have a drastic impact on the conclusions that are drawn from the model. The authors themselves mention in the text and describe in the SI that the key result that the sweep probability is independent of the mutation rate is highly dependent on the details of the growth process. In fact, ‘either a positive or a negative effect of mutation rate on sweep probability’ can be observed depending on the underlying growth dynamics. Cancers typically grow in highly heterogeneous environments, and exhibit different growth properties depending on the cell type of origin or metastatic site -- so this is likely a relevant perturbation to the model that can drastically change the results.

The sweep probability is independent of the mutation rate not only in our main model but also in the case of boundary-driven growth. In the case of exponentially expanding populations, we observe only a weak dependence on mutation rate (Figure 5, green curves). Only in the case of constant or tightly bounded population size, which is outside the scope of our study, do we observe a strong dependence on mutation rate. We have revised the relevant text (lines 315-317 and 325-332) to clarify these points. We conclude, “In summary, we find that the main conclusions drawn from our primary model also hold for alternative models of range expansions (exponential or boundary growth) but not for non-growing or tightly bounded populations.”

Indeed, if we look to the literature, it has also been shown that, in spatially structured models parametrized by cancer contexts, local measures of heterogeneity are limited due to cell turnover (cf, Waclaw, et al. ‘A spatial model predicts that dispersal and cell turnover limit intratumour heterogeneity’). This is in stark contrast to what the authors conclude -- that spatial structure increases heterogeneity. My understanding is that this is due in part to the fact that the present authors primarily focus on the sweep probability (a global measure of loss of heterogeneity). However, in a spatial context, spatially continuous measures of heterogeneity might be more appropriate to use and to compare, at least over the relevant observation timescales.

The question of how spatial structure influences tumour evolutionary mode and intratumour heterogeneity was the focus of our previous study (Noble *et al.*, *Nature Ecology & Evolution*, 2022). In that paper we used extensive agent-based simulations to examine not only a version of the model studied by Waclaw *et al.* (2015) but also models with alternative spatial structures, including the one we use in the current study. Our 2022 paper specifically discusses how results obtained using the agent-based model we use in the current study relate to the results of Waclaw *et al.* (2015). Consistent with Waclaw *et al.* (2015) and other prior studies, we broadly concluded that spatial structure inhibits selection and thus increases intratumour heterogeneity, and that, among spatial models, those with higher cell turnover and higher cell dispersal permit more selection and generate lower intratumour heterogeneity. We have also previously used spatial agent-based modelling to investigate how selective sweeps can purge intratumour heterogeneity (Noble &

Burley *et al.*, *Evolutionary Applications*, 2020). The scope of the current work is to obtain analytical results that provide general, mechanistic insights into one important aspect of the evolutionary dynamics observed in these prior studies: the frequency of selective sweeps. We expect that our mathematical analysis could be extended to provide insights into the dynamics of intratumour heterogeneity but this is beyond the scope of the current manuscript. We have added to the Introduction (lines 63-69) and the Discussion (lines 456-459) to clarify these points.

Taken collectively, these issues indicate to me that the results are of somewhat restricted applicability, and are highly dependent on the particular, relatively simple formulation of the model.

As above, we have added extensive new results and revised the text to show that our conclusions also hold for alternative models.

Minor concern:

Please describe the "warlock" code in more detail, perhaps illustrating what its output looks like -- the manuscript would greatly benefit from having this be described in a self-contained way, even if the software was published elsewhere.

We have added plots showing example model output (Figure 4A-B, S7 and S8) and more details of the model (lines 500-508).

Reviewer #2 (Remarks on code availability):

The code is contained in a rather large compressed file (>4GB) -- it would be much more preferable to store the code relevant to the paper (which is presumably somewhat simple, and significantly smaller than 4GB), in a Github repository.

We have created such a GitHub repository and cited it in the Code Availability section.

Reply to reviewer 1

The authors have addressed all my comments, and I recommend the manuscript for publication.

We thank the reviewer again for helping us to improve our manuscript.

Reviewer #1 (Remarks on code availability):

The GitHub and Zenodo repositories are well structured and contain Readme files to guide readers. I appreciate that some integrations are verified and documented using Mathematica. Reproducing agent-based simulations is not easy, as it requires generating and analyzing data using the demon package from other publications.

For readers who want to re-run the simulations, we recommend using the automated computational workflow 'warlock' rather than the underlying demon C++ code. We now highlight this workflow and cite the relevant preprint (Bak *et al.*, *arXiv*, 2023) at the start of the relevant Methods section (lines 487-489). As explained in that preprint, "The warlock repository encapsulates demon into an automated and reproducible computational workflow to simplify parallel simulations and make the software accessible to a wider community. This workflow is implemented in Snakemake, one of the most popular workflow management systems adopted by the bioinformatics community. In addition to local execution, we use an inbuilt mechanism to provide an interface for parallel High Performance Computing via SLURM workload manager. Hundreds of demon instances with varied parameter values can thus be executed in parallel with a single operation. All dependencies are resolved by Anaconda. The analysis of model output can also be automated simply by adding further components to the pipeline."

Reply to reviewer 2

Stein et al have amended their manuscript in two primary ways, adding more simulation results and expanding introductory and discussion points. I appreciate that effort has been put in to address my points of concern. However, the presentation of some of the additional results (extending the simulation model to random and accumulative fitness affects) come across as confusing, and potentially incorrect. Moreover, the relationship of these new simulation results to the analytical model is somewhat tenuous. Finally, the key predictions of the model wrt its relationship to cancer data seem to be demonstrably inconsistent with observations described in the literature.

To address the reviewer's concern about confusing readers, we have thoroughly revised and simplified the presentation of the new simulation results. To clarify the relationship between the new simulation results and the analytical model, we have reanalyzed the data and obtained new analytical results. To explain better how our results are consistent with empirical cancer data, we have rewritten the relevant parts of the Discussion, in consultation with authors of one of the most pertinent papers.

Major points:

I will first address the new individual-based simulations (those pertaining to Fig 4 in the revised text). There are two new simulations that are introduced:

1. Simulations in which random effect size mutations are introduced on the WT background, but further mutations cannot accumulate.
2. Simulations in which mutations can continue to accumulate “multiplicatively.”

The simulations in which random effect single mutations are introduced are problematic conceptually. The study aims to describe sweep probabilities, which are typically infinite time quantities, but individual-based models can only be run for finite time (presumably 1000 generations in this study, according to figure 4A, though this is never stated). The precise time to see a sweep (and therefore, the appropriate time to run a simulation) will sensitively depend on several underlying parameters like mutation rate, the mode of overall population growth (fixed, boundary-drive, exponential), and the distribution of fitness effects. Therefore, it is not clear that the “macroscopic” model and analytical results derived from it are the appropriate point of comparison for finite time data generated by individual-based simulations in the case where mutation effect sizes vary. Indeed, if the simulations were run for longer, we would eventually see an atypically large effect size beneficial mutation arise and eventually sweep in every simulation by chance (assuming an exponential distribution of effect sizes as the authors state).

The reviewer is correct that, of necessity, we ran simulations for only a finite period whereas our analytical predictions are based on integrating over all times. As we now state in the Methods (lines 529-530), simulations were stopped at a population size of 1 million or at 2,000 generations (whichever was reached first). However, the reviewer’s intuition that an exceptionally fit mutant is eventually certain to sweep is incorrect.

First consider the models in which we restrict the number of mutations that can accumulate per individual. Typically, all individuals reach this maximum mutation limit within the simulation time span, so no mutations (and in particular no mutations with atypically large effect size) could occur after the simulation stop time.

In the model with unrestricted accumulation of mutations, if we were to use strictly multiplicative fitness effects (so that a mutation with effect s increases proliferation rate by a factor of $1 + s$) then evolution would be unconstrained. Fitter mutants would expand faster, proliferate faster, and so acquire ever more mutations in a positive feedback loop. The model would thus, within finite time, generate infinitely fit “Darwinian demons”. To prevent this physically impossible scenario, we instead assume in our simulations that evolution is constrained by diminishing returns epistasis, such that the proliferation rate of mutants cannot exceed ten times that of the wildtype. We describe this simulation feature in the Methods (lines 520-523; Equation 17). Due to diminishing returns epistasis, the fitness effect of new mutations generally diminishes as the population size grows.

Although the above arguments refute the claim that “we would eventually see an atypically large effect size beneficial mutation arise and eventually sweep in every

simulation”, the reviewer is correct that we might have observed more sweeps had we run the simulations for longer. To estimate the extent of such undercounting, we have conducted a new analysis of the population sizes at which detected sweeps occurred (Methods lines 532-539 and 556-573; new Figures S10-S14). This analysis shows that the distribution of population sizes is bell-shaped and is well approximated by a Gamma distribution. By fitting truncated Gamma distributions to the simulation results, we inferred the truncation points and hence the proportion of sweeps missing from each set of simulations. We have adjusted the data points in Figure 5C (previously Figure 4C) to account for the inferred numbers of missing sweeps. As we report in the Methods, “The adjustments are negligible except in the case of small fitness effects, for which sweep probabilities are in any case low. In tests using artificially truncated distributions from simulations with large s values (new Figure S15), we find that this method typically overestimates the proportion of missing sweeps. The adjusted values can therefore be regarded as upper bounds.” Our main results and conclusions are unaffected.

Moreover, the “anomaly” at low fitness observed in these simulations and described in Figure 4C is confusing, and I’m not sure the explanation provided in the text is satisfactory. Specifically, what is even being measured in Figure 4C? The probability of the first mutation sweeping or the probability of any mutation sweeping? What is the x-axis precisely? Figure 4D is more clear wrt the x-axis, but it shows that the sweep probability increases as more mutations are allowed to accumulate, which is a more realistic scenario than the analytical model where mutations do not accumulate.

To make a meaningful comparison between the results of the fixed-effect and random-effects models, we need to use an appropriate summary statistic of the distribution of mutation fitness effects. As we now state in the Results (lines 291-293), the problem is that “the random fitness effect determines not only a mutant’s expansion speed but also its probability of evading stochastic extinction” and hence determines the effective mutation rate. As we now state in the Methods (lines 547-550), “For example, in a Moran process with $K = 16$, nearly 94% of nearly-neutral mutations succumb to stochastic extinction before they can contribute to the macroscopic evolutionary dynamics, compared to only 50% of mutations that confer fitness effect $s = 1$ ”. Hence, it is inappropriate to compare results simply in terms of the mean fitness of all generated mutations. We chose the x-axes in the previous Figures 4C and 4D after considering several less satisfactory alternatives. In response to the reviewer’s comment we have revisited this problem and arrived at a better solution.

As now explained in the Methods (lines 550-553), “To account for this difference in effective mutation rates, we calculated the mean effect of mutations that reached an abundance of at least 10 individuals in the at-most-one-mutation model (see Figure S16 for the distributions).” Considering in this way only the “contending” mutations that escape stochastic extinction is conventional (see review by Bataillon & Bailey 2014) and is consistent with previous studies (e.g. Gerrish & Lenski 1998; Ralph & Coop 2010). As recorded in the Acknowledgements, we discussed this approach with a leading expert on distributions of mutant fitness effects, Guillaume Martin of the Institut des Sciences de l’Évolution de Montpellier, who agreed it is appropriate to

our study. We have rewritten the relevant sections of the Results (lines 286-323) and Methods (lines 540-573) accordingly.

With this simpler summary statistic on the x-axis, we are confident that the new Figure 5C is easier to understand than the previous Figures 4C and 4D. As we report in the Results (lines 296-298), when compared in terms of this better summary statistic, “sweep probabilities in random-fitness-effect simulations are close to the predictions obtained under the assumption of fixed effects”. For a given distribution of fitness effects, the sweep probability increases as more mutations are allowed to accumulate (in line with intuition).

Although we believe that the new Figure 5C is the best way to present our results, we also think it is useful to show that the similarity between sweep probabilities in random-fitness-effect and fixed-effect simulations holds for an alternative choice of x-axis. We have therefore retained the previous Figure 4C as Supplementary Figure S5B and added a variant of this plot (without pooling results across parameter sets) in Figure S5A. As now mentioned in the Results (lines 323-326), Figure S5 also shows that “sweeps are more often due to a single large-effect mutation than several mutations with small effects”. We have rewritten the figure caption to answer the reviewer’s questions regarding the axes.

We have also retained Section 9 of the SI Text, which discusses the “anomaly” at low fitness in Figure S5B. We believe SI Text Section 9 is useful because it sheds light on the subtleties inherent to summarising and interpreting results when mutation effects are random; because it describes an effect that other researchers may observe when analysing similar results; and because the transition matrix analysis lays the groundwork for further mathematical investigations. We have added a two-paragraph introduction to this section of the SI Text (lines 347-360).

For the simulations that allow for multiplicative accumulation of mutations, the problems are potentially more severe. According to the process by which the proliferation rate of mutations changes (Eq. 17), for some of the values stated (for instance, $s=0.1$) the proliferation rate decreases on average (this can be seen by simulating the process described by Eq. 17). Since further mutations are relatively deleterious on average in this model, it is perhaps not so surprising that the fixation of mutant lineages is somewhat limited.

We are grateful to the reviewer for spotting a typo in Eq. 17. A closing parenthesis was misplaced so that the equation in the text differed from that in the simulation code.

We wrote

$$r_{m,i} = \min \left\{ r_{m,i-1} (1 + s) X \left(1 - \frac{r_{m,i-1}}{M} \right), M \right\}$$

It should be

$$r_{m,i} = \min \left\{ r_{m,i-1} \left(1 + s X \left(1 - \frac{r_{m,i-1}}{M} \right) \right), M \right\}$$

In the correct equation, all mutations are beneficial. Importantly, all simulations were based on the correct equation (as can be verified by reviewing the history of the code in the GitHub repository).

Moreover, simulations with different fitness scales s are apparently pooled to generate Figs 4C-E and may have contributed to the "anomaly" described in the text and above.

As we now explain in the SI Text (lines 359-360), the spike anomaly "is not simply due to the pooling of results because it also appears in single-batch results", as shown in new Figure S5C. We have revised the caption of the former Fig 4C (now Fig S5B) to clarify that it shows pooled results.

In the new Figures 5C, 5D and S5A, each data point corresponds to one batch of simulations with identical parameter values. There is no pooling of results and no spike anomaly at low fitness values.

Finally, I will highlight a key prediction that the authors bring up, that is seemingly untrue. The statement "we predict...detectable tumours will almost always contain multiple competing clones with different driver mutations" (line 360) is central to the message of the paper (and I appreciate that the authors were bold enough to go out on a limb and say this). Unfortunately, it is likely false. The authors cite a convenient figure from Dentro et al. (that 95% of tumors in PCAWG exhibit subclonal expansion, line 429) but neglect the key observation from that same paper that only 11% of subclones carry a driver. This suggests that subclonal dynamics are driven by considerations other than purely fitness on a fixed landscape per se (for instance, there could perhaps be 'ecological' effects or a deformed fitness landscape). In an ecological scenario, for instance, sweeps would be prevented by more "trivial" mechanisms like disjoint niche preferences. Beyond that, it is clear that there is evidence for "multi-hit" origins of certain cancers, which involve the acquisition of multiple clonal mutations and therefore imply selective sweeps. If the authors were to provide some hard evidence from cancer data to back up some of the key predictions of the model, I would love to see it.

We appreciate that the quoted final sentence of the final section of our Results could reasonably be interpreted as going out on a limb to propose an unconventional hypothesis. This was not our intention. We have therefore rephrased the sentence (lines 357-360) to provide a more accurate and precise synopsis: "In summary, our macroscopic model (assuming equal mutation effects and no accumulation of mutations) predicts that selective sweeps during a clonal expansion are rare unless mutations are very strongly beneficial, in which case sweeps begin and end early in the expansion."

We fully agree that "subclonal dynamics are driven by considerations other than purely fitness". As we now clarify at the start of the Results (lines 82-84), "Our macroscopic model is designed to test whether clonal interference alone can prevent selective sweeps, and to obtain upper bounds on selective sweep probabilities during range expansions." In a new Discussion paragraph (lines 424-440), we explain the ways in which microenvironmental heterogeneity is likely to thwart selective sweeps. We also give other reasons why the general conclusion that "selective sweeps during clonal expansions are restricted to strong driver mutations during the early stages of expansion" might be expected to hold even when driver mutation effects are random and multiplicative. We also now explain (on lines 473-476) that, "To obtain more precise predictions rather than upper bounds, it will be

important to examine how microenvironmental heterogeneity, immune responses, and phenotypic plasticity inhibit clonal expansions in each cancer type.”

As the reviewer suggests that we might have misinterpreted or misrepresented the findings of Dentro *et al.* (*Cell*, 2021), we sought advice from two of the co-first authors of that paper: Jonas Demeulemeester and Maxime Tarabichi (who are also the postdoc co-supervisors of Alex Stein, the first author of our paper). As now mentioned in the Acknowledgments, these two researchers read our manuscript, answered our questions about their study, and offered suggestions to help us rephrase relevant passages. As the reviewer points out, not all clonal expansions in Dentro *et al.* can be directly attributed to driver mutations. However, the 11% mentioned by the reviewer is a conservative lower bound for the following reasons: (i) The 11% refers only to known driver mutations. As shown by Gerstung *et al.* (*Nature*, 2020) and as also argued by Dentro *et al.*, driver mutations that are typically found to be subclonal arise from a more diverse pool of genes than driver mutations that are typically found to be clonal. This suggests we have likely not yet identified all possible driver mutations. (ii) The 11% refers to SNVs and indels only. The analysis excluded CNAs and SVs that can also act as driver mutations. (iii) Positive dN/dS ratios suggest widespread positive selection of ‘functional’ mutations, further indicating the involvement of genetic mutations in clonal expansions. The complete quote from Dentro *et al.* acknowledges the first two limitations: “Our observations highlight a gap in knowledge about drivers of subclonal expansions; only 11% of all subclones carry a known SNV or indel driver mutation. Late tumor development could be driven largely by different mechanisms (such as CNAs and SVs [Jamal-Hanjani *et al.*, 2017; Mamlouk *et al.*, 2017] or epigenetic alterations), or the pool of late driver mutations is too large to have been sampled extensively enough so far.” Regarding the third factor: “Analysis of dN/dS ratios revealed positive selection across subclones and cancer types. Although our analyses do not exclude the possibility that a fraction of tumors evolve under weak or no subclonal selection, they support widespread selection.” We have revised and expanded the relevant parts of the Discussion (lines 441-452) to clarify these points.

Our results are entirely consistent with “multi-hit” cancer origins. In the Discussion, we now refer explicitly to “multi-stage models” and explain how recent experimental studies have tried to estimate the time of driver events with partial success. To lessen the risk of misunderstanding, we have revised the part of the Discussion that explains how clonal driver mutations can arise through bottlenecks (lines 453-473).

In summary: With regard to cancer, our methods are designed to establish an upper bound on the selective sweep probability during a clonal expansion. Since we want an upper bound, we omit from the model factors that would impede selective sweeps, such as microenvironmental heterogeneity and accelerating propagation speed. As we find that the upper bound is low, we conclude that multiple clonal driver mutations observed in tumours most likely arise not during the final clonal expansion but instead through population bottlenecks in a stepwise expansion process. We are not proposing an unorthodox hypothesis but rather determining which previously proposed explanation is the most likely, thus helping to resolve a debate in the field (as discussed in our Introduction, lines 53-70, and Discussion, lines 453-457).

Minor points:

I appreciate that a version of the code has been uploaded to Github. It appears that much of the underlying simulation code resides in a single >4000 line c++ file (https://github.com/robjohnnoble/demon_model/blob/master/src/demon.cpp). Considering that previous versions of this code have ostensibly been reviewed elsewhere, reviewing this is beyond my scope. However, this file is somewhat difficult to navigate and could stand to be more thoroughly commented or broken up into more easily parsed chunks. Moreover, I was not able to get it to run.

As the reviewer points out, the simulation framework 'demon' was central to previous peer-reviewed papers (Noble & Burley *et al.*, *Evolutionary Applications*, 2020; Noble *et al.*, *Nature Ecology & Evolution*, 2021) and has therefore been scrutinized before. We appreciate that working with C++ code can be challenging, which is why we created the automated computational workflow 'warlock' to make our software more accessible. For readers who want to rerun the simulations, we recommend using warlock rather than the underlying demon C++ code. We now mention this workflow at the start of the relevant Methods section (lines 487-489) and cite a preprint that describes how it works and how to use it (Bak *et al.*, *arXiv*, 2023).

Without details of the reviewer's software environment and the error messages they encountered, we are unable to comment on why they were unable to run our code. We and others have successfully run the code on Windows, macOS and Unix operating systems on personal computers and on the computing clusters of three universities.